# Identification of Compounds of *Crocus sativus* by GC-MS and HPLC/UV-ESI-MS and Evaluation of Their Antioxidant, Antimicrobial, Anticoagulant, and Antidiabetic Properties

**DOI:** 10.3390/ph16040545

**Published:** 2023-04-05

**Authors:** Aziz Drioiche, Atika Ailli, Nadia Handaq, Firdaous Remok, Mohamed Elouardi, Hajar Elouadni, Omkulthom Al Kamaly, Asmaa Saleh, Mohamed Bouhrim, Hanane Elazzouzi, Fadoua El Makhoukhi, Touriya Zair

**Affiliations:** 1Research Team of Chemistry of Bioactive Molecules and the Environment, Laboratory of Innovative Materials and Biotechnology of Natural Resources, Faculty of Sciences, Moulay Ismaïl University, B.P. 11201 Zitoune, Meknes 50070, Morocco; 2Medical Microbiology Laboratory, Mohamed V. Hospital, Meknes 50000, Morocco; 3Research Team of Enhancement and Protection of Plants, Laboratory of Environmental Biology and Sustainable Development, Higher Normal School, Abdelmalek Essaadi University, Tetouan 93000, Morocco; 4Department of Pharmaceutical Sciences, College of Pharmacy, Princess Nourah Bint Abdulrahman University, P.O. Box 84428, Riyadh 11671, Saudi Arabia; 5Laboratory of Biological Engineering, Team of Functional and Pathological Biology, Faculty of Sciences and Technology Beni Mellal, University Sultan Moulay Slimane, Beni-Mellal 23000, Morocco

**Keywords:** *Crocus sativus* L., phorone, crocins, picrocrocin, safranal, DPPH, FRAP, total antioxidant capacity, antimicrobial activity, anticoagulant activity, antidiabetic activity

## Abstract

In order to valorize the species *Crocus sativus* from Morocco and to prepare new products with high added value that can be used in the food and pharmaceutical industry, our interest was focused on the phytochemical characterization and the biological and pharmacological properties of the stigmas of this plant. For this purpose, the essential oil of this species, extracted by hydrodistillation and then analyzed by GC-MS, revealed a predominance of phorone (12.90%); (R)-(-)-2,2-dimethyl-1,3-dioxolane-4-methanol (11.65%); isopropyl palmitate (9.68%); dihydro-*β*-ionone (8.62%); safranal (6.39%); *trans*-*β*-ionone (4.81%); 4-keto-isophorone (4.72%); and 1-eicosanol (4.55%) as the major compounds. The extraction of phenolic compounds was performed by decoction and Soxhlet extraction. The results of the determination of flavonoids, total polyphenols, condensed tannins, and hydrolyzable tannins determined by spectrophotometric methods on aqueous and organic extracts have proved the richness of *Crocus sativus* in phenolic compounds. Chromatographic analysis by HPLC/UV-ESI-MS of *Crocus sativus* extracts revealed the presence of crocin, picrocrocin, crocetin, and safranal molecules specific to this species. The study of antioxidant activity by three methods (DPPH, FRAP, and total antioxidant capacity) has proved that *C. sativus* is a potential source of natural antioxidants. Antimicrobial activity of the aqueous extract (E_0_) was investigated by microdilution on a microplate. The results have revealed the efficacy of the aqueous extract against *Acinetobacter baumannii* and *Shigella* sp. with MIC ≤ 600 µg/mL and against *Aspergillus niger*, *Candida kyfer*, and *Candida parapsilosis* with MIC = 2500 µg/mL. Measurements of pro-thrombin time (PT) and activated partial thromboplastin time (aPTT) in citrated plasma obtained from routine healthy blood donors were used to determine the anticoagulant activity of aqueous extract (E_0_). The anticoagulant activity of the extract (E_0_) studied showed that this extract can significantly prolong the partial thromboplastin time (*p* < 0.001) with a 359 µg/mL concentration. The antihyperglycemic effect of aqueous extract was studied in albino Wistar rats. The aqueous extract (E_0_) showed strong in vitro inhibitory activity of α-amylase and α-glucosidase compared with acarbose. Thus, it very significantly inhibited postprandial hyperglycemia in albino Wistar rats. According to the demonstrated results, we can affirm the richness of *Crocus sativus* stigmas in bioactive molecules and its use in traditional medicine.

## 1. Introduction

The overproduction of reactive oxygen species generates damage in biomolecules (DNA, proteins, amino acids, and lipids) and can produce oxidative stress, which subsequently can induce various pathologies, such as neurodegenerative diseases, cardiovascular diseases, diabetes, and cancer [1,2]. Currently, there are therapies against these various pathologies, but they present serious side effects, most often undesirable. To reduce these side effects, several works have focused on the research of effective bioactive molecules from natural plant resources, in particular aromatic and medicinal plants.

As humankind has been progressing in its development, we have developed interactions with plants to take better advantage of them for our well-being. We have empirically learned some of the virtues of plants and have improved them with the development of science. However, plants in their natural state continue to be used by most of the world’s population. In particular, the Moroccan population benefits from a rich and diversified flora characterized by high endemism of 7000 existing species, subspecies, and varieties with 940 genera and 135 families of which 537 are endemic to the country and 1625 are rare or threatened [3]. Its diversity is the backbone of Morocco’s industry for aromatic and medicinal plants, which is regarded as the world’s 12th largest exporter of these plants.

The Fez-Meknes area, with its rich and varied ecosystems, offers an ideal setting for the research of new bioactive molecules of agro-food and pharmaceutical interest. Saffron is one of the plants of this region which plays a very important role at the economic and social level, and its conservation constitutes a major and preponderant priority.

The species *Crocus sativus* has been widely used as a dyeing agent, as well as for medication in the traditional medicinal systems of India, Thailand, the Philippines, Malaysia, China, Australia, Africa, etc. [4,5]. The Moroccan flora is also abundant in this species, which is used as a condiment plant and in traditional medicine to treat a variety of illnesses [6].

Saffron (*C. sativus* L.) is an autumnal herbal flowering plant of the family Iridaceae. It is currently ranked as the most expensive spice in the world. It is the most important plant of the genus *Crocus*. With an estimated 418 tons produced per year, this plant is commonly cultivated in Iran, Morocco, India, Afghanistan, Italy, Greece, and Spain [7]. The most commonly used plant parts of *C. sativus* are the stigmas, flowers, and petals. Saffron has many traditional applications, and it is used to treat dysmenorrhea, gastric ulcers [8], nervous system disorders, asthma, whooping cough, and inflammations [9]. It can be used as an aperitif, stimulant, and tonic [8]. In addition, phytochemical studies of saffron have identified multiple active components such as *cis*- and *trans*-crocin (dye), safranal (aroma), and glycoside carotenoids [10]. In addition, the petals and stigmas of this plant contain various flavonoids, phenylpropanoids, phenolic glycosides, alkaloids, coumarins, terpenoids, fatty alcohols, and fatty acids of which glycosylated flavonoids and anthocyanins are considered important antioxidant and anti-inflammatory compounds [11,12,13].

Currently, the literature on saffron provides a deep scope on plant varieties, major constituents, and culinary uses, mainly based on traditional Chinese, Indian, and Iranian medicine. To enlarge the scope of the future implications of this Moroccan plant species for medicinal and culinary purposes, the present study aims to valorize *C. sativus* cultivated in the Boulemane area, through phytochemical investigations and evaluation of biological and pharmacological properties of this plant species. Certainly, the phytochemical profile, antioxidant, antimicrobial, and anticoagulant properties, and antidiabetic activities of the stigmas of *C. sativus* cultivated in the Boulemane area are highlighted by taking into account in vitro and in vivo investigations.

## 2. Results

### 2.1. The Yield of C. sativus EO

The average yield of EO was calculated based on the dry plant material obtained from the stigmas of *C. sativus*.

The yield of the obtained essential oil is very low, about 0.25 ± 0.01% (Table 1). This oil is characterized by a yellowish color with an aromatic spicy and woody smell.

### 2.2. GC-MS Analysis of C. sativus Essential Oil

The GC/MS analyses of EO from *C. sativus* stigmas were used to determine the chromatographic profile (Figure 1), identify the different constituents, and measure their relative abundances in the EO analyzed (Table 2).

The analysis of the chemical composition of the EO identified 27 chemical compounds for *C. sativus* representing 99.99% of the total composition. By examining the EO’s major chemical classes (Figure 1), we found that the saffron sample presents an EO rich in oxygenated monoterpenes (67.97%), oxygenated sesquiterpenes (13.35%), hydrocarbon diterpenes (7.23%), and oxygenated diterpenes (7.01%). The hydrocarbon sesquiterpenes and hydrocarbon monoterpenes come last, with levels of 3.11 and 1.32%, respectively. In addition, the majority of compounds of this EO are phorone (12.90%); (R)-(-)-2,2-dimethyl-1,3-dioxolane-4-methanol (11.65%); isopropyl palmitate (9.68%); dihydro-*β*-ionone (8.62%); safranal (6.39%); *trans*-*β*-ionone (4.81%); 4-keto-isophorone (4.72%); and 1-eicosanol (4.55%).

The EO of *C. sativus* is composed essentially of ketones (44.28%) and non-aromatic alcohols (18.66%), followed by esters (12.06%), hydrocarbons (11.66%), and aldehydes (7.64%). In addition, we discovered additional chemical families, such as weakly proportioned phenols and epoxides, that describe the species of saffron (Figure 2).

### 2.3. Phytochemical Screening

Phytochemical tests showed that *C. sativus* stigmas contain important secondary metabolites (Table 3). A strongly positive reaction was observed showing the presence of sterols and triterpenes, flavonoids, gallic tannins, alkaloids, and saponosides. In addition, a negative reaction of anthocyanins, catechic tannins, and anthracene derivatives was observed.

### 2.4. Extraction and Quantitative Analysis of Phenolic Compounds

#### 2.4.1. Extraction Yields

We performed the solid–liquid extraction by decoction (E_0_), by Soxhlet water (E_1_), and by Soxhlet ethanol–water (E_2_) extraction, to compare the yields and the contents of phenolic compounds obtained by these two methods. From Figure 3, it appears that the yield of extracts obtained by decoction for *C. sativus* stigmas is higher than those obtained by Soxhlet extraction.

#### 2.4.2. Determination of Phenolic Compounds

In order to evaluate the contents of polyphenols, flavonoids, condensed tannins, and hydrolyzable tannins in the aqueous extracts and the hydroethanolic extract of *C. sativus*, we established calibration curves for gallic acid (Y = 0.095X + 0.003; R^2^ = 0.998), quercetin (Y = 0.073X − 0.081; R^2^ = 0.995), catechin (Y = 0.7421X + 0.0318; R^2^ = 0.998), and tannic acid (Y = 0.1700X − 0.0006718; R^2^ = 0.996). The quantities of total polyphenols, flavonoids, condensed tannins, and hydrolyzable tannins in the extracts were estimated by milligram equivalents of gallic acid, quercetin, vanillin, and tannic acid per gram of extract, respectively.

The results of the total polyphenol assay show that the contents of these molecules vary considerably from one extract to another (Figure 4). The best contents recorded for *C. sativus* stigmas were noted for the hydroethanolic extract (45.474 mg EAG/g), followed by decocted and then the aqueous extract obtained by Soxhlet extraction. Flavonoid content values (Figure 4) show that *C. sativus* extracts are visibly rich in flavonoids with the best contents recorded for the hydroethanolic extract (33.497 mgEQ/g), followed by decocted (113.877 mgEQ/g) then the aqueous extract obtained by Soxhlet extraction (5.639 mgEQ/g). An analysis of the tannin results shows that *C. sativus* stigmas are rich in tannins (condensed and hydrolyzable tannins). The extracts obtained by Soxhlet extraction of *C. sativus* represent the highest tannin contents, while the decocted one has a lower content of condensed tannins and hydrolyzable tannins (0.558 mg EC/g and 0.364 mg EAT/g extracts).

#### 2.4.3. Analysis and Identification of Polyphenols in *C. sativus* Extract by High-Pressure Liquid Chromatography–Mass Spectrometry (HPLC/UV-ESI-MS)

The extracts of *C. sativus* were analyzed by HPLC/UV-ESI-MS, and the chromatogram in Figure 5 presents the different compounds detected in the stigmas of *C. sativus*. The analysis of the mass spectra in addition to the chromatogram allowed us to propose 27 molecules which are recorded in Table 4.

The analytical study of the mass spectra of the aqueous and hydroethanolic extracts of *C. sativus* in negative mode shows the presence of several carotenoids, flavonoids, and phenolic acids. The principal carotenoids identified in the decocted extract (E_0_), aqueous extract obtained by Soxhlet extraction (E_1_), and hydroethanolic extract obtained by Soxhlet extraction (E_2_) are *cis*-crocin-3 (0.32%, 1.64%, and 1.48%), *cis*-crocin-4 (1.06%, 2.58%, and 4.12%), crocetin (0.57%, 0.16%, and 2.47%), crocin-2 (7.8%, 11.25%, and 24.58%), crocin-3 (25.65%, 10.1%, and 3.58%), crocin-4 (10.52%, 1.32%, and 17.99%), picrocrocin (18.78%, 30.23%, and 0.5%), *trans*-crocin-1 (0.62%, 2.34%, and 4.3%), and *β*-carotene (1.32%, 4.44%, and 7%), respectively. We also recorded the presence of safranal in the three extracts (E_0_, E_1_, and E_2_) with respective percentages in the range of 0.78%; 3.83%; and 0.69%.

Each compound can be mass fragmented using electrospray mass spectrometry (ESI-MS) detection in the negative ion mode, which is an additional technique for determining their structure.

The ESI-MS spectrum of picrocrocine (C_16_ H_26_ O_7_, *m/z* = 330) showed an addition ion (with formic acid (HFA)) at *m/z* = 375 [M-H + HFA]^−^.

The crocin-4, crocetin esterified with a gentiobiose at each end, *trans* and *cis* isomers (calculated as C_44_ H_64_ O_24_, M = 976) produced the pseudo-molecular ion [M-H]^−^ at *m/z* = 975 at 25.19 and 26.96 min, respectively. Upon adduct formation with formic acid in the mass spectrum of crocin-4, the peak formed with *m/z* = 1021 was identified as [M-H + HFA]^−^.

The crocin-3 (C_38_ H_54_ O_19_, M = 814), crocetin esterified with a gentiobiose at one end and glucose at the other end, *trans* and *cis* isomers showed the [M-H]^−^ ion at *m/z* = 813. The use of formic acid as a mobile phase modifier gave an intense signal at *m/z* = 859 corresponding to the presence of adduct ion with crocin-3 [M-H + HFA]^−^.

The crocin-2 (C_33_ H_44_ O_14_, M = 652), a crocetin esterified with a gentiobiose at one extreme and a free carboxyl group at the other, showed the [M-H]^−^ ion at *m/z* = 651 at 26.18 min. The compound has ion fragmentation similar to that of crocin-4 and crocin-3.

The *β*-carotene showed the [M-H]^−^ ion at *m/z* = 535 at 26.27 min. On adduct formation with formic acid in the mass spectrum of *β*-carotene, the peak formed with *m/z* = 581 was identified as [M-H + HFA]^−^.

The other ions at *m/z* = 489 [M-H]^−^ correspond to the pseudomolecular ion of *trans*-crocin-1 and at *m/z* = 327 [M-H]^−^ to the quasi-molecular ion of crocetin. Thus, the pseudomolecular ions at *m/z* = 149 [M-H]^−^ and *m/z* = 409 [M-H]^−^ correspond to the molecules of safranal and oleanolic acid, respectively.

### 2.5. Heavy Metal Contents

The determination of heavy metal contents in the *crocus* genus has been the subject of very little work. In our study, a total of seven elements (As, Fe, Cd, Sb, Cr, Pb, and Ti) were determined. The *Crocus sativus* sample showed values below the FAO/WHO regulated limit values (Table 5).

### 2.6. Antioxidant Activity

Three techniques—DPPH, FRAP, and TAC—were used to assess the antioxidant properties of the aqueous and hydroethanolic extracts of C. sativus and the reference (ascorbic acid). The calibration curves of ascorbic acid by DPPH (Y = 1.013X − 8.032; R^2^ = 0.9893), FRAP (Y = 0.004760X + 0.09740; R^2^ = 0.8963), and TAC (Y = 0.04066X + 0.02110; R^2^ = 0.9949) methods were determined.

The extracts are referred to as natural antioxidants, due to their ability to reduce and/or prevent free radical formation. The results in Figure 6A show that the aqueous extracts and the hydroethanolic extract of *C. sativus* are endowed with antiradical power. The hydroethanolic extract (E_2_) and the aqueous extract obtained by Soxhlet extraction (E_1_) exerted remarkable antioxidant activity with EC_50_ values equal to 565.590 and 627.003 μg/mL, respectively. The EC_50_ of ascorbic acid was 19.378 μg/mL. The EC_50_ of iron reduction in the *C. sativus* extracts is quite evident, according to the results of the FRAP method. Figure 6B shows significant differences between the reducing power of the extracts from *C. sativus* stigmas and the positive control (ascorbic acid). The aqueous extract obtained by Soxhlet extraction of *C. sativus* showed a better reducing capacity than the decocted extract (112.530 µg/mL) or the aqueous extract obtained by Soxhlet extraction (113.914 µg/mL). However, these results fall short of the ascorbic acid standard, which had a concentration of 0.470 g/mL. As the total antioxidant activity (TAA) is measured in terms of ascorbic acid equivalents, the phosphomolybdenum technique is quantitative. According to Figure 6C, the tested extracts showed good total antioxidant capacity. The results showed that the organic extracts presented the highest TAC compared to the decocted extract, with a high activity recorded for the aqueous extract (193.356 mg EAA/g) followed by the hydroethanolic extract (175.666 mg EAA/g). Moreover, the results grouped in Figure 7 present a perfect correlation between the phenolic compound contents of the studied extracts and the different antioxidant activities carried out, in particular between the contents of flavonoids, condensed tannins, and hydrolyzable tannins concerning DPPH, FRAP, and TAC assays.

### 2.7. Antimicrobial Activity

The results of the antimicrobial activity of decocted *C. sativus* extract are represented in Table 6. The MIC of the extract was categorized using the standards put forth by Sartoratto, Duarte, Wang, Oliveira, and their associates [14,15,16,17]. Antimicrobial activity was classified as high (MIC < 600 μg/mL), moderate (MIC between 600 and 2500 μg/mL), and low (MIC > 2500 μg/mL). The MIC, BMC, and FMC analyses of *C. sativus* show a high bactericidal quality of the decocted extract against enterobacterial species. The other strains tested show some resistance. Indeed, the evaluated decocted extract is powerful against Gram-negative bacilli and, in particular, against *A. baumannii* and *Shigella* sp. with MIC lower than or equal to 600 µg/mL. Moreover, the decocted extract of *C. sativus* is more active against strains causing candidiasis, in particular against *C. kefyr* and *C. parapsilosis* with a MIC of about 2500 µg/mL. The mold (*A. niger*) was also sensitive to the decocted extract studied (MIC = 2500 µg/mL).

### 2.8. Anticoagulant Activity

According to the data in Figure 8, it is apparent that the studied extract (E_0_) does not affect prothrombin time (PT). However, the results of the coagulation test (aPPT) showed a remarkable anticoagulant activity by inhibiting the intrinsic pathway in a dose-dependent manner. Indeed, *C. sativus* extract (E_0_) can significantly prolong the partial thromboplastin time (*p* < 0.001) with a 359 µg/mL concentration.

### 2.9. Antidiabetic Activity

#### 2.9.1. Evaluation of the Inhibitory Effect of Decocted Extract on the Activity of α-Amylase and α-Glucosidase, In Vitro

Regarding the in vitro inhibition tests of α-amylase and α-glucosidase by aqueous extract of *C. sativus*, the results presented in Figure 9 show that the extract exerts a variable and dose-dependent inhibitory effect. *C. sativus* decocted extract as well as acarbose showed strong inhibitory activity on α-amylase and α-glucosidase. The effect of acarbose against α-amylase and α-glucosidase was increased in a concentration-dependent manner, with EC_50_ of 364.446 and 17.269 μg/mL, respectively. The decocted extract’s effect was significantly higher than that of acarbose on α-amylase and α-glucosidase (86.326 mg/mL; 10.957 µg/mL, respectively).

#### 2.9.2. Acute Toxicity Study of *C. sativus* Decocted Extract

The result of this acute toxicity assay shows that the decocted material is not toxic even at 2 g/kg. Neither toxicity (diarrhea, vomiting, abnormal mobility, etc.) nor mortality were caused by the decocted substance during the observation period.

#### 2.9.3. Study of the Antihyperglycemic Activity of *C. sativus* Decocted Extract in Normal Rats In Vivo

Analysis of the glucose tolerance test and comparison of the total areas under the blood glucose curve during the 150 min period are shown in Figure 10 and Figure 11.
Oral glucose tolerance testBlood glucose levels in normal rats showed a high peak 30 min after glucose loading. A positive effect on the response of rats to glucose loading was noted in rats treated with decocted extract and glibenclamide. Oral administration of *C. sativus* extract at a dose of 400 mg/kg, 30 min before glucose overload, to normal rats significantly attenuated postprandial hyperglycemia for this decocted extract study (Figure 10), compared with the group of control rats pretreated with distilled water. However, glibenclamide very significantly inhibited postprandial hyperglycemia during the first hour (60 min) after glucose overload, (*p* < 0.001; 1.08 g/L) compared with the distilled water-pretreated rat group.Areas under the curve (AUCs) of postprandial glucose levels.The area under the curve was significantly smaller in the decocted extract-treated rats (*p* ≥ 0.001; 56.11 g/L/h) than in the distilled water-treated rats (62.91 g/L/h). In addition, the area under the curve of glibenclamide was significantly (*p* ≥ 0.01) smaller (55.95 g/L/h) compared with the area under the curve of distilled water-treated rats (62.91 g/L/h) (Figure 11).

## 3. Discussion

More than 100 volatile chemical compounds are found in saffron after processing the scent precursors and dehydrating the stigmas [18]. Monoterpenes and sesquiterpenes obtained from the isoprenoid synthesis pathway, phenylpropanoids and benzenoids derived from the shikimic acid pathway, and chemicals derived from the enzymatic conversion of lipids by *β*-oxidation are the several types of aromatic compounds found in saffron. A few volatile chemicals are also created by shortening or altering the skeletons of other molecules [19]. Safranal, which makes up about 0.001–0.006% of the dry matter and is responsible for 30–70% of the saffron’s scent, is the main constituent [20]. Picrocrocine is converted into safranal through an enzyme reaction and/or dehydration. However, there is proof that this conversion can take place at low pH levels.

The yield obtained in EO from *C. sativus* stigmas in our work remains low and comparable to the yields of saffron harvested in Mexico by Cid-Pérez [21] and in Kashmir, Iran, and Turkey by Kafi [22]. Previous studies on the chemical composition of *C. sativus* EO show similarities and variations in major constituents. Using GC-MS/FID, Anastasaki et al. [23] identified the following significant chemicals in samples of *C. sativus* from Italy and Spain: safranal, 4-hydroxy-2,6,6-trimethyl-3-oxocyclohexa-1,4-diene-1-carboxaldehyde, isophorone, and dihydrooxophorone. The same substances, albeit in a different sequence of concentration, were found in samples from Iran and Greece. Aliakbarzadeh et al. [24] used GC-MS to identify 77 volatile compounds of Iranian saffron, of which 10 are biomarkers, incuding 8 secondary metabolites (isophorone; phenylethyl alcohol; phorone; keto-isophorone; dihydrooxophorone; safranal; 2,6,6-trimethyl-4-oxo-2-cyclohexene-1-carbaldehyde; 2,4,4-trimethyl-3-carboxaldehyde-5-hydroxy-2,5-cyclohexadiene-1-one), 1 is a primary metabolite (linoleic acid), and 1 is a long chain fatty alcohol (nonacosanol). Azarabadi and Özdemir [25] used solid phase microextraction (SPME) to extract and determine the significant compounds from Iran, including acetic acid; 2-(5H)-furanone; isophorone; 4-keto-isophorone; 2,6,6-trimethyl-1,4-cyclohexanedione; eucarvone; and safranal. Similarly, Carmona et al. [26] determined high concentrations of acetic acid, dihydrooxophorone, and isophorone in samples from Morocco and Iran as markers to distinguish their origin. This polymorphism of the identified constituents varies according to the time of harvesting, seasonal factor, place of origin, and the condition of the plant, fresh or dry. Thus, the concentration of compounds of interest may vary according on the preparation, extraction, and characterization processes [24].

The results of phenolic compound assays indicated that the content of phenolic compounds in the studied extracts showed a significantly higher variation in the aqueous extract and hydroethanolic extract obtained by Soxhlet extraction compared to the decocted extract. The total contents of phenolic compounds, flavonoids, and tannins in *C. sativus* stigmas were higher compared to those reported by the work of Karimi et al. in Iran [27]. Additionally, this work offers the first account of a comparison of three distinct extracts made via decoction, Soxhlet extraction, and ethanol/water mixture. These data confirm that *C. sativus* stigmas harvested from the Boulemane region are rich in total polyphenols, flavonoids and tannins, and could therefore show important biological and pharmacological activities. In addition, the results of the chromatogram and mass spectral analysis (HPLC/UV-ESI-MS) obtained were in agreement with earlier studies in a significant way [28]. The provided information offers trustworthy confirmation of the mass spectral analysis-based structures of crocins, picrocrocin, *β*-carotene, crocetin, and safranal. Mineral elements are involved in important biological functions of cells. The contents of heavy metals (As, Cd, Cr, Fe, Pb, Sb, and Ti) determined in *C. sativus* powder were found to be below the standards described by the FAO/WHO. This will allow safe and healthy uses of the studied plant in the fields of pharmacy, food, and cosmetics.

Concerning the evaluation of the antioxidant potential of *C. sativus* extracts, we measured the antioxidant activities using three methods (DPPH, FRAP, and TAC). The results of these activities showed the richness of the studied extracts of *C. sativus* in antioxidant molecules. Several previous works have focused on the antioxidant effects of *C. sativus* flower, petal, and leaf extracts, however, we found that few studies have been reported on the antioxidant activity of stigma extracts [29,30,31]. The outcomes of our research demonstrate that the antioxidant potential of the *C. sativus* stigma extracts from the Boulemane region is comparable to the information gathered by Assimopoulou et al. [32], who presented studies on the antioxidant activity of saffron. They estimated that the antioxidant activity of saffron could be attributed to two bioactive compounds, crocin and safranal. According to Chen et al. [33], who compared the antioxidant activity of Chinese saffron (*C. sativus*) with that of Gardenia jasminoides, Chinese saffron had greater free radical scavenging abilities than Gardenia, with corresponding values between 107 and 421 mg tocopherol/g. In the present work, the antioxidant activity of saffron stigma extracts could be attributed to the presence and synergic effects of phenolic compounds (total polyphenols, flavonoids, and tannins) or other bioactive molecules, as affirmed by the correlation test between antioxidant activities and phenolic compound contents described in Figure 7. Moreover, the difference between the antioxidant activity of the extracts studied in this work and those from China could be due to the difference in their polyphenol, flavonoid, and tannin contents and experimental conditions (plant, standard, etc.). Indeed, many studies have previously mentioned that phenolic compounds have remarkable antioxidant properties in both in vitro and in vivo systems [34,35,36].

Up to now, some global studies have reported work on the antimicrobial activity of *C. sativus* extracts. Jadouali and co-workers demonstrated that *C. sativus* leaf extracts had strong antibacterial activity against *Listeria* sp. in a prior investigation carried out in Morocco [29]. Vahidi and his team investigated the antimicrobial efficacy of *C. sativus* extracts prepared with various solvents and harvested in Khorasan, Iran, against various bacterial and fungal strains [37]. Asgarpanah and co-workers studied the antibacterial properties of methanolic extract made from the various portions of *C. sativus* that were harvested from the southern Iranian province of Khorasan. They focused on the extract’s ability to inhibit different types of bacteria and fungi [38]. Aqueous and ethanolic extracts of *C. sativus* against mammitis pathogens in Turkey showed an intriguing antibacterial action [39]. Additionally, Muzaffar demonstrated the potent antibacterial properties of *C. sativus* extracts made from India-sourced methanol and petroleum ether [40]. Furthermore, our findings demonstrated that *C. sativus* decocted extract is superior to Gram-positive bacteria in its ability to combat *Aspergillus niger*, candidiasis, and Gram-negative pathogens. This difference could be due to the difference in the cell wall structure of the tested microorganisms. Moreover, our results showed that the antibacterial activity of *C. sativus* extracts from Morocco is similar to those from Iran and Turkey [37,38].

By forming a platelet thrombus, the normal hemostatic process halts a cut or wound on the blood vessels; once healing is complete, the thrombus is eventually removed. This complex multiphasic mechanism depends on platelets and coagulation factors interacting with blood arteries. Thrombosis or hemorrhage can result from a flaw in any of these stages [41]. In this study, the decocted extract showed a clear difference between coagulation and anticoagulation by prolonging the partial thromboplastin time (aPPT) in a more significant manner. Earlier research with saffron demonstrated that crocin might inhibit thrombosis in rats, delay blood clotting time and relieve respiratory discomfort during pulmonary thrombosis in mice, and decrease platelet aggregation in rabbits [42]. Hence, crocetin effectively inhibited platelet aggregation caused by collagen and ADP, but not by arachidonic acid [43]. Furthermore, while neither platelet adherence to collagen nor cyclic AMP levels were impacted by crocetin, it greatly reduced the release of dense granules [44]. After this, administration of saffron tablets (200 and 400 mg/day) failed to significantly affect the coagulation and anticoagulation system after one month in a double-blind, placebo-controlled clinical investigation with a large sample size. The authors suggested that the case reports of bleeding complications could be due to the high dose of saffron, the long period of consumption, or idiosyncratic activities [45].

The results of the antidiabetic activity of the decocted extract studied against digestive enzymes showed higher inhibitory activities against α-glucosidase and α-amylase compared to acarbose and compared to other published studies [46]. The traditional usage of *C. sativus* aqueous extract in the treatment of diabetes is supported by its capacity to inhibit the enzymes α-glucosidase and α-amylase, which suggests that substances with antidiabetic activity are extracted into the water. Since limiting the rate of glucose absorption from the intestines into the bloodstream was thought to be the only method that could help prevent diabetes, numerous prior studies have shown the hypoglycemic effect of phenolic compounds, improving postprandial blood glucose, acute insulin secretion, and insulin sensitivity [47].

## 4. Materials and Methods

### 4.1. Vegetal Material

The studied sample of *Crocus sativus* was collected from the cultivated stands in the Boulemane region. The data on the origin, the part collected, and the site of harvesting are shown in Table 7. The Scientific Institute of Rabat’s Laboratory of Botany and Plant Ecology is where identification of this species was completed.

### 4.2. Microbial Materials

The determination of the antimicrobial activity of the aqueous extract of *C. sativus* was performed on twenty-four bacterial and eight fungal strains (Table 8). These particular microorganisms are harmful and well-known for their strong resistance, ability to invade, and toxicity toward humans. They are frequently encountered in many infections in Morocco that represent a clinical and therapeutic problem. These strains were isolated from the hospital environment: Mohamed V-Meknes Provincial Hospital. All strains were kept in a 20% glycerol stock at −80 °C, rejuvenated in Mueller–Hinton and Sabouraud broths, and subcultured before use.

### 4.3. Animal Selection for Research

The acute toxicity study was performed on albino mice (male and female). The study of the antidiabetic activity in vivo was carried out on Wistar rats (males and females). The animals were kept in an environment with a photoperiod of 12 h of light and 12 h of darkness and a temperature of 22 ± 2 °C in the Biology Department’s animal home at the Faculty of Sciences in Oujda. The animals were maintained under favorable rearing conditions with free access to water and food. The animals were cared for, used, and handled in full compliance with internationally accepted standard guidelines and the institutional animal ethics committee (02/2019/LBEAS) [48].

### 4.4. The Qualitative and Quantitative Study of Essential Oils

#### 4.4.1. Extraction of Essential Oils from *C. sativus* and Determination of Yield

Essential oils (EOs) were extracted from *C. sativus* stigmas by hydrodistillation using a Clevenger-type apparatus. Briefly, 20 g of plant material was boiled for three hours with 200 mL of distilled water to produce three distillations. Then, the obtained oil was dried by adding anhydrous sodium sulfate (Na_2_SO_4_) and stored at a temperature of −4 °C in a dark bottle until its use. The yield of EO was calculated from 20 g of the plant material by Formula (1) [49]:(1)$$Yield\;(\%)=\frac{W\;(EO)}{W0}\times 100$$
where: W (EO): weight of HE recovered (g); W0: weight of plant material (20 g).

#### 4.4.2. Analysis and Identification of the Chemical Composition of *C. sativus* EO

The sample of the studied essential oil was subjected to chromatographic analysis using a gas chromatograph of the Thermo Electron (Trace GC Ultra) type in conjunction with a mass spectrometer of the Thermo Electron Trace MS system (Thermo Electron: Trace GC Ultra; Polaris Q MS). The fragmentation was accomplished by an electron impact of intensity of 70 eV. The chromatograph was outfitted with a flame ionization detector (FID) fed by an H2/air–gas mixture and a DB-5 type column (5% phenyl-methyl-siloxane) (30 m × 0.25 mm × 0.25 m film thickness). The column temperature was set to increase by 4 °C/min for 5 min, from 50 to 200 °C. The carrier gas utilized was nitrogen at a flow rate of 1 mL/min in the split injection mode (leak ratio: 1/70).

The identification of the chemical composition of the EO was performed by determining and comparing the Kovats indices (KIs) of the compounds with those of the known standard products described in the databases of Kovats [50], Adams [51], and Hübschmann [52]. The retention times of the peaks were compared to those of known authentic standards kept in the authors’ lab, and their reported KI and MS data were compared to those kept in the WILEY and NIST 14 standard mass spectral database and published literature to identify every compound. This was carried out using the Kovats index. The retention times of any products were compared using Kovats indices to the retention times of linear alkanes with the same number of carbons. They were determined by co-injecting a mixture of alkanes (standard C_7_-C_40_) under the same operating conditions.

### 4.5. Phytochemical Screening

In this qualitative investigation, chemical families were discovered using tests for the solubility of compounds, precipitation, and turbidity responses. It can also be carried out by looking for a specific color change or by carrying out an examination under UV light. The stigmas of *C. sativus* were used for this phytochemical investigation. Dry samples of the plant were ground into a fine powder, followed by characterization tests of different chemical groups, carried out according to the protocols of Dohou et al., Judith, Mezzoug et al., Bekro et al., Bruneton, and N’Guessan et al. [53,54,55,56,57,58].

### 4.6. Study of Phenolic Compounds

#### 4.6.1. Extraction of Phenolic Compounds

Extraction of phenolic compounds was performed using two methods: decoction and solid–liquid extraction by Soxhlet apparatus. The first sample of 30 g was added to 600 mL of distilled water, heated, and boiled at 80 °C for 1 h. The mixture was allowed to sit for five minutes before being filtered at a lower pressure. The decocted extract was dried in an oven at 70 °C and then recovered as a powder in a glass vial and stored until use. The Soxhlet apparatus was used to extract the second and third samples, each weighing 30 g, using 300 mL of water or an ethanol/water solution (70/30, *v*/*v*) as the extraction solvent. After different extraction cycles, the extracts were concentrated using a rotary evaporator. Table 9 below describes the coding adopted for the extracts prepared in this work.

#### 4.6.2. Determination of Total Polyphenols

The Folin–Ciocalteu method, as described by Singleton and Rossi [59], was used to determine the total polyphenol content of the plant extracts under study. The methods of their determination are generally based on the oxidation of these compounds and the development of color. The Folin–Ciocalteu technique is the most popular. It consists of the reduction of a mixture of phosphotungstic (H_3_PW_12_O_40_) and phosphomolybdic (H_3_PMo_12_O_40_) acids (Folin–Ciocalteu reagent) into a mixture of blue oxides of tungsten (W_8_O_23_) and molybdenum (Mo_8_O_3_). The assay of the compounds is carried out by colorimetry with an optical density reading. Using a spectrophotometer (UV mini-1240) set to 760 nm, the absorbance reading was taken in comparison to a blank (a reaction mixture without extract). Gallic acid was used as a positive control from a concentration range of 50 µg/mL in a parallel calibration curve that was made under the same operating circumstances. The results are expressed as the milligram equivalent of gallic acid per gram of extract (mg GAE/g) and were determined by the equation of type Y = a x + b obtained with the calibration curve. Each test was repeated three times.

#### 4.6.3. Determination of Flavonoids

The flavonoid content was determined by the colorimetric method with aluminum trichloride by adaptation of the methods of Djeridane [60] and Hung [61] and their co-workers. Aluminum chloride (AlCl_3_) forms a complex with the hydroxyl groups (OH) of flavonoids. The flavonoids were estimated by UV spectroscopy at a wavelength of 433 nm. Quercetin, a standard flavonoid that experienced the same analytical conditions as the samples and had concentrations between 5 and 30 µg/mL, was used to generate a calibration curve (type Y = a x + b) for the measurement of flavonoids. The flavonoid content is expressed as milligram equivalent of quercetin per gram of extract (mg EQ/g). Each test was repeated three times.

#### 4.6.4. Determination of Condensed Tannins

Condensed tannin contents were estimated by the vanillin method [62]. Briefly, different prepared concentrations of (+)-catechin solution (2 mg/mL) were added to a volume of 3 mL of vanillin/methanol solution (4%; *m*/*v*). The mixture was manually stirred. Then, 1.5 mL of strong hydrochloric acid was added to each concentration. The resulting mixes were allowed to react for 20 min at room temperature. With the aid of a UV–visible spectrophotometer, the absorbance at 499 nm was determined in comparison to a blank. By substituting our samples for the catechin in the calibration curve plotting process, the quantity of condensed tannins in our samples was determined. The calibration curve was used to quantify the tannin concentration in milligram equivalent of catechin per gram of dry matter weight.

#### 4.6.5. Determination of Hydrolyzable Tannins

Hydrolyzable tannins were determined by the method of Willis and Allen [63], with minimal modifications. The extract was vortexed for ten seconds with 5 mL of 2.5% KIO_3_ in a mixture of 10 µL of the extract. After 2 min for the extract and the reaction’s optimum, and 4 min for the standard tannic acid solution, the maximum absorbance was reached. A spectrophotometer was used to measure the absorbance at 550 nm (UV–visible). The results are expressed as mg of tannic acid per gram of dry plant, and 11 different tannic acid concentrations (ranging from 100 to 2000 g/mL) were used to elaborate the calibration curve.

#### 4.6.6. HPLC/UV ESI-MS Analysis of *C. sativus* Stigma Extracts

Analysis of phenolic compounds of *C. sativus* decocted by high-performance liquid chromatography coupled to Q Exactive Plus mass spectrometry with electrospray as a molecular ionization method (HPLC/UV-ESI-MS) was performed with an UltiMate 3000 HPLC (Thermo Fisher Scientific, Sunnyvale, CA, USA) equipped with a sample changer, in which the samples were stored at 5 °C. This HPLC system was equipped with a reverse phase C_18_ column (250 × 4 mm, id 5 μm, Lichro CART, Lichrospher, Merck, Darmstadt, Germany). During the analysis, the column temperature was set at 40 °C. The mobile phase degassed by ultrasonic treatment was: solvent A: 0.1% formic acid in water (*v*/*v*) and solvent B: 0.1% formic acid in acetonitrile (*v*/*v*). The gradient composition was: 2% B at the beginning (0 min) and then changed to 30%, 95%, 2%, and 2% B at 20, 25, 26, and 30 min, respectively. The flow rate was 1 mL/min and the injection volume was 20 μL.

Detection was performed on a Maxis Impact HD (Bruker Daltonik, Bremen, Germany) in MS/MS mode (broadband collision-induced dissociation (bbCID)) after negative electrospray ionization. In addition, UV detection by an L-2455 diode array detector (Merck-Hitachi, Darmstadt, Germany) was performed by scanning in the wavelength range of 190–600 nm and then at three acquisition wavelengths, 280 nm, 320 nm, and 360 nm. The values of the other parameters were set as follows: capillary voltage of 3000 V; drying gas temperature 200 °C; dry gas flow rate 8 L/min; a nebulizing gas pressure of 2 bar; and an offset plate of −500 V. Nitrogen was used as the desolvation gas and nebulizer gas. MS data were acquired over an *m/z* range of 100 to 1500. A Thermo Scientific™ Chromeleon™ 7.2 Chromatography Data System (CDS) was used as software for data acquisition and evaluation. The analysis of the eluted compounds was carried out by analyzing the mass spectra of the eluted molecules.

### 4.7. Heavy Metal Assays: Inductively Coupled Plasma Atomic Emission Spectrometry (ICP-AES)

A number of heavy metals, including arsenic (As), cadmium (Cd), chromium (Cr), iron (Fe), lead (Pb), antimony (Sb), and titanium (Ti), were studied. Each metal has a set of somewhat lax contamination requirements. Drugs whose primary ingredients are known to accumulate significant amounts of cadmium are exempt from this rule. The standard mineralization protocol (AFNOR, 1999) using aqua regia (HNO_3_ + 3 HCl) was chosen for the analysis of the concentrations of the key elements (As, Cd, Cr, Fe, Pb, Sb, and Ti). The latter permits large sample sizes, which reduce issues with sample representativeness. The method involves mixing the pulverized plant material (0.1 g) with 3 mL of aqua regia, which is made from 1 mL of nitric acid (HNO_3_; 99%) and 2 mL of hydrochloric acid (HCl; 37%). The mixture was then placed in a reflux setup at 200 °C for two hours, following which it was cooled and allowed to settle. The supernatant was then taken, filtered through a 0.45 m membrane, and diluted to a final volume of 15 mL using distilled water. The ICP-AES (Ultima 2 Jobin Yvon) at the UATRS laboratory (Technical Support Unit to Scientific Research) at CNRST in Rabat measured the concentrations of heavy metals [64].

### 4.8. Antioxidant Activities

#### 4.8.1. Antiradical Activity by the DPPH^•^ Test

The evaluation of the antiradical activity was based on the ability of an antioxidant (phenolic compound) to donate a single electron to the synthetic radical DPPH^•^ (chemical compound 2,2-diphenyl-1-picrylhydrazyl), with purple coloring, to stabilize it into DPPH with yellow coloring [65]. At a wavelength of 515 nm, the experiment was carried out in a UV–visible spectrophotometer. The 6 × 10^−5^ M DPPH^•^ solution was obtained by dissolving 2.4 mg of DPPH^•^ in 100 mL of ethanol. In order to prepare the extract samples, pure ethanol was used to dissolve them. The test was carried out by combining 200 µL of extract (sample) or standard antioxidant (ascorbic acid) at various concentrations with 2.8 mL of the preceding DPPH^•^ solution.

Following 30 min of incubation at room temperature in the dark, the absorbance was measured at 515 nm in comparison to a blank that solely contained ethanol. The DPPH^•^ solution without extract served as the negative control, and the values were subsequently converted into percent inhibition using the formula below (2) [66]:(2)$$\%\;AA=\frac{Abs\;control-Abs\;sample}{Abs\;control}\times 100$$
where: % AA: percentage of antiradical activity; Abs control: absorbance of the blank (optical density of the solution consisting of DPPH^•^ and ethanol); Abs sample: absorbance of the test compound (extracts).

#### 4.8.2. Ferric Reducing Antioxidant Power (FRAP) Method

According to Oyaizu’s approach from 1986, the reduction of Fe^3+^ in the K_3_Fe(CN)_6_ complex to Fe^2+^ was used to assess the iron reducing activity of our extracts. To carry out the measurements of antioxidant activity by the FRAP method, samples from the different extracts were studied, proceeding as described below. A quantity of extract underwent the same treatment as the samples described above. Using distilled water to calibrate the device, the absorbance value was read at 700 nm against a prepared blank (UV–Vis spectrophotometer). A typical antioxidant solution was used as the positive control (ascorbic acid whose absorbance was measured under the same conditions as the samples). Increases in absorbance were accompanied by increases in the extracts’ tested reducing power [67,68].

#### 4.8.3. Total Antioxidant Capacity (TAC)

The TAC of the extracts was evaluated by the phosphomolybdenum method described by Khiya [69]. According to this method, molybdenum Mo (VI), which is present as molybdate ions MoO_4_^2−^, is reduced to molybdenum Mo (V) MoO^2+^ in the presence of the extract to form a green phosphate/Mo(V) complex at an acidic pH. Each extract was combined with 3 mL of the reagent solution and a 0.3 mL aliquot (0.6 M sulfuric acid, 28 mM sodium phosphate, and 4 mM ammonium molybdate). The tubes were secured with screws and left to sit at 95 °C for 90 min. Following cooling, the solutions’ absorbance at 695 nm was measured in comparison to a produced blank that was incubated under identical circumstances as the sample. Several ascorbic acid concentrations were prepared as a standard range. Ascorbic acid milligram equivalents per gram of crude extract (mg EAA/g EB) are used to express the TAC.

### 4.9. Determination of the Minimum Inhibitory Concentration, Minimum Bactericidal Concentration, and Minimum Fungicidal Concentration

Using 96-well microplates and the reference microdilution method, the minimum inhibitory concentration (MIC) was determined [70]. The MIC is the lowest amount of EO necessary to completely stop the tested microorganism’s development during incubation, as measured by how much growth is visible to the naked eye. Thus, a series of dilutions were carried out from a stock solution of the essential oil produced in 10% DMSO to obtain concentrations of 5 to 0.93 × 10^−2^ mg/mL of each EO. These dilutions were made with a final volume of 100 µL for each concentration in Sabouraud broth for fungi and Mueller–Hinton medium for bacteria. Thereafter, 100 µL of microbial inoculum with a final concentration of 10^6^ or 10^4^ CFU/mL for bacteria or fungi, respectively, was added to the various stages of the dilution series. Ten microliters of resazurin was added to each well as a measure of bacterial growth after a 24 h incubation period at 37 °C. The coloring changed from purple to pink after a second incubation at 37 °C for two hours, indicating microbial development. The MIC value was determined as the lowest concentration that prevents resazurin from changing color. The growth and sterility controls, respectively, were the eleventh and twelfth wells. For this oil, the test was conducted twice. Terbinafine 250 mg, the typical antifungal used in the study, was mixed with 2 mL of 10% DMSO after being ground. The minimum bactericidal concentration (MBC) and minimum fungicidal concentration (MFC) were determined by taking 10 µL from each well that had no apparent growth and plating it for 24 h at 37 °C on Mueller–Hinton (MH) agar for bacteria or in Sabouraud broth for fungi. The lowest sample concentration that resulted in a 99.99% reduction in CFU/mL relative to the control was designated as the MBC and MFC. Additionally, it was possible to calculate the MBC/MIC or MFC/MIC ratio of each extract to evaluate its antimicrobial potency. Accordingly, if the ratio is less than 4, the essential oil has a bactericidal/fungicidal effect, and if it is greater than 4, the sample has a bacteriostatic/fungistatic effect [71].

### 4.10. Anticoagulant Activity

The anticoagulant effect was evaluated by chronometric coagulation tests involving the prothrombin time and partial thromboplastin time by adopting the method described by Hmidani et al. [72]. The prepared decocts were studied for possible investigation of anticoagulant agents. The extract concentrations used in the coagulation mixtures were 11.500, 5.750, 2.875, 1.438, 0.719, 0.359, and 0.179 mg/mL. In a polypropylene container, 3.8% trisodium citrate tubes were used to collect blood samples. They were then separated and pooled to create a plasma pool after being immediately centrifuged at 25,000 rpm for 10 min. Before use, the freshly made plasma pool was kept at −10 °C.

By combining the citrated normal plasma pool (50 µL) with a plant extract solution (50 µL) and incubating for 10 min at 37 °C, the partial thromboplastin time (aPTT) of the various samples examined was determined. After that, the mixture was mixed with 100 µL of the PTT reagent (CKPREST^®^) provided by Diagnostica Stago and incubated for 5 min at 37 °C. Coagulation was induced by adding 25 mmol/L CaCl_2_ (100 µL) and the coagulation time was recorded. In contrast, the prothrombin time (PT) experiment was performed by combining 50 µL of the citrated normal plasma pool with 50 µL of a plant extract solution, and incubating the mixture for 10 min. After that, 200 µL of Neoplastin ^®^ Cl reagent was added and preincubated for 10 min at 37 °C. The clotting time was then noted. The anticoagulant activity of the various aqueous plant extracts was measured in seconds at various doses. A coagulometer (MC4Plus MERLIN Medical^®^) was used to automatically carry out each measurement six times.

### 4.11. Antidiabetic Activity

#### 4.11.1. Study of the Inhibitory Effect of Aqueous Extracts on the Activity of Pancreatic α-Amylase, In Vitro

The following concentrations of acarbose were used for testing: 1, 0.8, 0.6, 0.4, and 0.2 mg/mL. The following concentrations were used to test the aqueous extract: 0.89, 0.45, 0.22, 0.11, and 0.06 mg/mL. The inhibitory effect of the aqueous extract on α-amylase enzymatic activity was measured according to the method described by Daoudi et al. [73]. Phosphate buffer solution was mixed with 200 µL of the aqueous extract solution or 200 µL of the acarbose solution (positive control). All tubes received an addition of 200 µL of the enzyme solution, with the exception of the blank tube, which received 200 µL of phosphate buffer instead. The tubes underwent a 10 min preincubation at 37 °C. Starch solution was then added to each tube in a volume of 200 µL. The entire set was incubated for 15 min at 37 °C. The tubes were filled with 600 µL of DNSA to halt the enzymatic process.

The tubes were then submerged for 8 min in a pot of boiling water. Heat shock was then used to inhibit this process. Before adding 1 mL of diluted water to each tube, the tubes were submerged in an ice water bath. Using a spectrophotometer and a blank made up of the buffer solution rather than the enzyme solution, the absorbance was measured at 540 nm. The following Equation (3) was used to compute the % inhibition of each extract or acarbose:(3)$$\%\;Inhibition=\frac{A\;control-A\;sample}{A\;control}\times 100$$
where: A control: absorbance of enzyme activity without inhibitor; A sample: absorbance of enzymatic activity in the presence of extract or acarbose.

#### 4.11.2. Study of the Inhibitory Effect of Aqueous Extracts on the Activity of α-Glucosidase, In Vitro

The α-glucosidase inhibitory activity of the extract was determined using pNPG substrate according to the method described by Chatsumpun et al. [74], with some modifications. The extracts were tested using the concentration range from 0.488 to 100 µmg/mL. DMSO (5%) was used to prepare each sample, and phosphate buffer was used to prepare the α-glucosidase enzyme (pH 6.8). As a solvent control, DMSO (5%) was employed, while acarbose served as the positive control. Step by step, 40 µL of the α-glucosidase enzyme (0.1 U/mL) and 10 µL of each sample were added to a 96-well plate. This mixture was then preincubated for 10 min at 37 °C. Thereafter, 50 µL of pNPG (1 mM) was added, and the mixture was incubated at 37 °C for 20 min. To end this reaction, 100 µL of Na_2_CO_3_ (0.1 M) solution was added. In a microplate reader, the mixture’s absorbance was determined at a wavelength of 405 nm. Equation (3) was used to compute the percentage of α-glucosidase inhibition.

#### 4.11.3. Acute Toxicity Study

This experiment aimed to demonstrate that normal mice do not experience any short-term toxicity from the therapeutic dose. We investigated acute toxicity by the oral pathway, as this is the usual pathway implicated under normal conditions for humans. This study was conducted according to the guidelines of the Organization for Economic Cooperation and Development (OECD) [75]. The products tested were *C. sativus* extract (E_0_) selected for the pharmacological study, at doses of 0.5, 1, and 2 g/kg.

Two lots of albino mice (20–35 g) in a fasted state (14 h) were randomly divided into 4 groups (n = 6; ♂/♀ = 1):Control: distilled water (10 mL/kg).Group 1: aqueous extract E_0_ (0.5 g/kg).Group 2: aqueous extract E_0_ (1 g/kg).Group 3: aqueous extract E_0_ (2 g/kg).

The mice were weighed before the test began. They were then given a single dose of the aqueous extract right away. After that, we continuously observed them for 10 h to look for indications of apparent toxicity. The mice were observed every day for the following 14 days for any new clinical or behavioral indications of harm.

#### 4.11.4. Study of the Antihyperglycemic Activity of the Aqueous Extract of *C. sativus* in Normal Rats In Vivo

The ex vivo oral glucose tolerance test (OGTT) or oral sucrose tolerance test (OSTT) was performed by administering the different test products to normoglycemic mice. This study aimed to investigate whether the extract (E_0_) had a postprandial antihyperglycemic effect in normal rats overloaded with D-glucose.

The normal rats (200–250 g) in a fasted state (14 h) were grouped into 3 groups (n = 6; ♂/♀ = 1):Control: administration of distilled water (10 mL/kg).Extract: administration of aqueous extract E_0_ (2 mL/kg).Glib: administration of glibenclamide (2 mg/kg).

First, normal rats were anesthetized with ether (inhalation), then a blood sample was taken from the queue to measure blood glucose at t_0_ and, immediately after that, the test material (distilled water, aqueous extract, or glibenclamide) was administered orally. Thirty minutes afterward, another blood glucose measurement was performed, and immediately afterward the rats were overloaded with D-glucose (2 mg/kg). Thereafter, the change in blood glucose was monitored for 3 h, at 30, 60, 90, and 150 min.

### 4.12. Statistical Analysis

The results were expressed as mean ± standard error of mean. The results were analyzed using the one-factor ANOVA test of variance and followed by the Tukey posttest using GraphPad Prism 9 software (version 9.5.1) (San Diego, CA, USA). Values with *p* < 0.05 are considered significant. Correlations between phenolic compound contents and antioxidant activities were investigated using R software (version 4.1.3).

## 5. Conclusions

Morocco is a country with abundant plant resources, and there is a wide variety of medicinal plants there that are used to treat and prevent a number of diseases. From this study, we conclude that essential oils and extracts of *Crocus sativus* stigmas from the Boulemane region contain the same bioactive molecules required by European and American norms for the export of this spice, knowing that Morocco became the 3rd world producer of saffron in 2021, in front of Greece and India. The studied extracts are rich in phenolic compounds and carotenoids. Moreover, the extracts of *Crocus sativus* stigmas studied have great antioxidant, antimicrobial, anticoagulant, and antidiabetic properties. As a result, in addition to its numerous uses in the food and medical industries, it can be employed as a source of bioactive molecules in the treatment of infectious disorders brought on by bacteria that are multidrug resistant.

## Figures and Tables

**Figure 1 pharmaceuticals-16-00545-f001:**
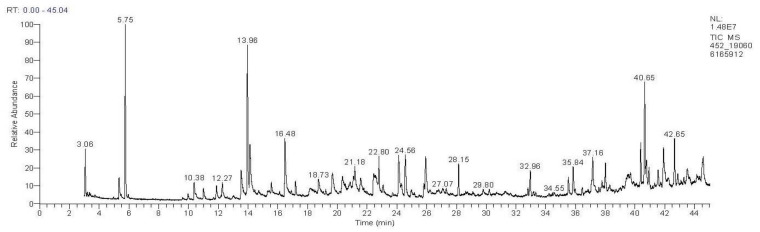
Chromatographic profile of the GC/MS analysis of *C. sativus* EO studied.

**Figure 2 pharmaceuticals-16-00545-f002:**
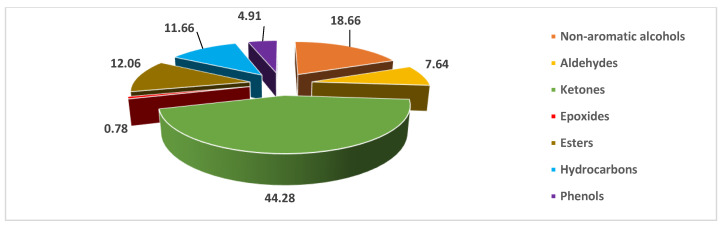
Distribution of chemical classes identified in *C. sativus* EO (%).

**Figure 3 pharmaceuticals-16-00545-f003:**
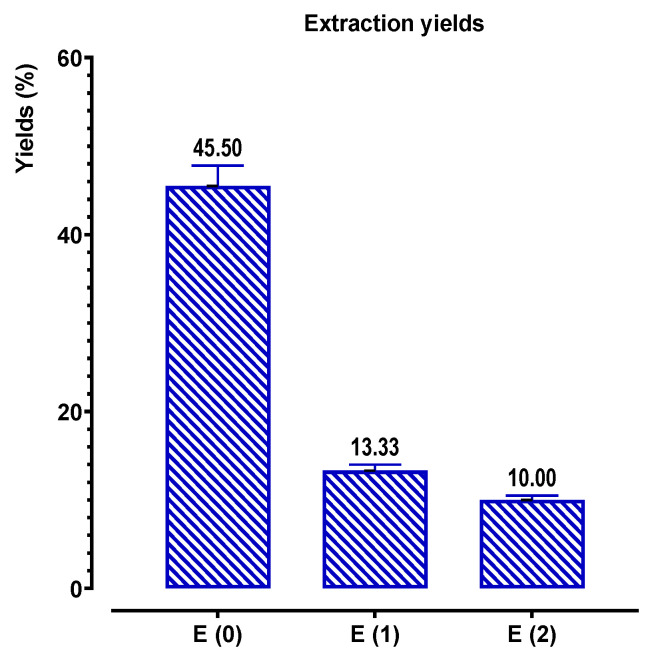
Extraction yields of phenolic compounds from *C. sativus*.

**Figure 4 pharmaceuticals-16-00545-f004:**
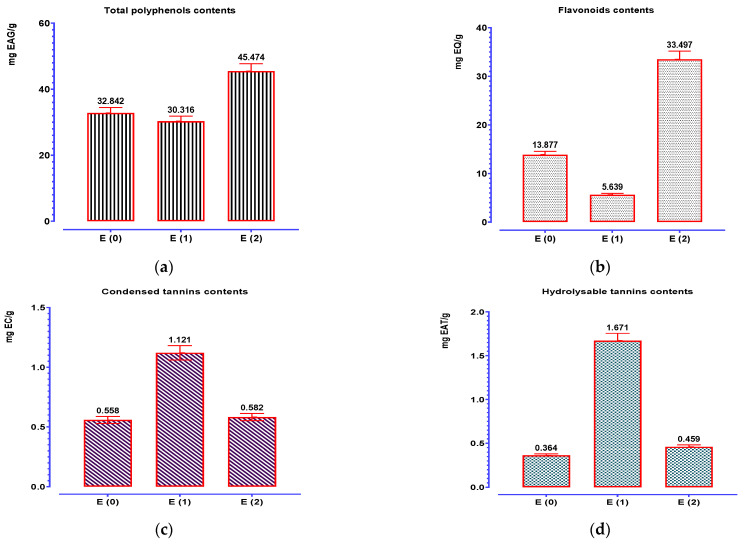
Contents of phenolic compounds: (**a**) total polyphenols; (**b**) flavonoids; (**c**) condensed tannins; (**d**) hydrolyzable tannins. Mean values ± standard deviations of determinations performed in triplicate are reported. Means are significantly different (*p* < 0.001).

**Figure 5 pharmaceuticals-16-00545-f005:**
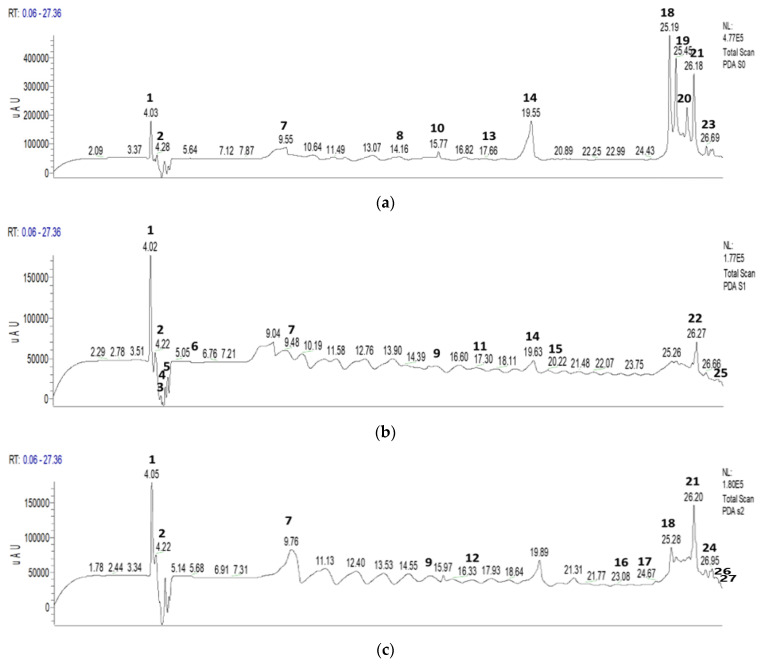
HPLC chromatograms of *C. sativus* compounds: (**a**) decocted; (**b**) aqueous extract obtained by Soxhlet extraction; (**c**) hydroethanolic extract obtained by Soxhlet extraction.

**Figure 6 pharmaceuticals-16-00545-f006:**
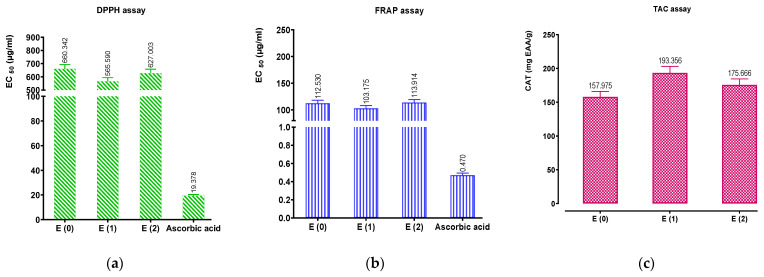
Antioxidant activity of ascorbic acid and extracts by: (**a**) DPPH; (**b**) FRAP; (**c**) TAC assays. Mean values ± standard deviations of determinations performed in triplicate are reported. Means are significantly different (*p* < 0.001).

**Figure 7 pharmaceuticals-16-00545-f007:**
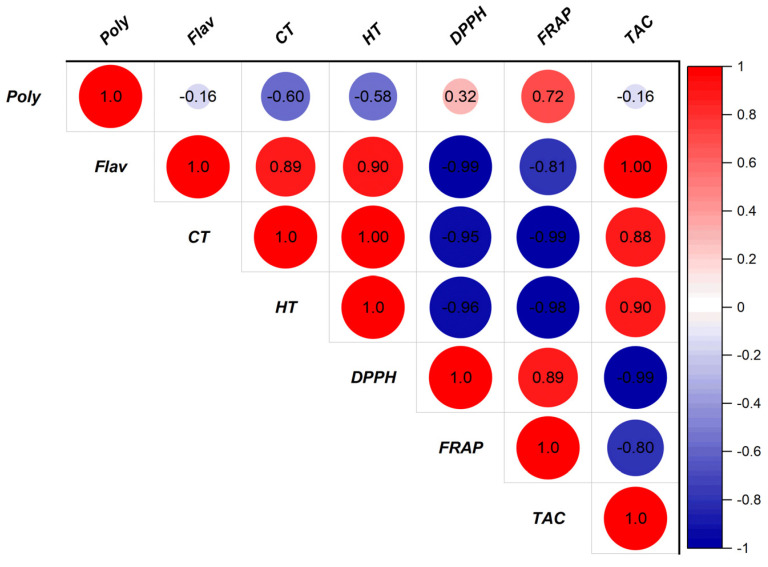
Correlation between antioxidant activities and phenolic compound contents of *C. sativus* stigma extracts. (Poly: polyphenols; Flav: flavonoids; CT: condensed tannins; HT: hydrolyzable tannins).

**Figure 8 pharmaceuticals-16-00545-f008:**
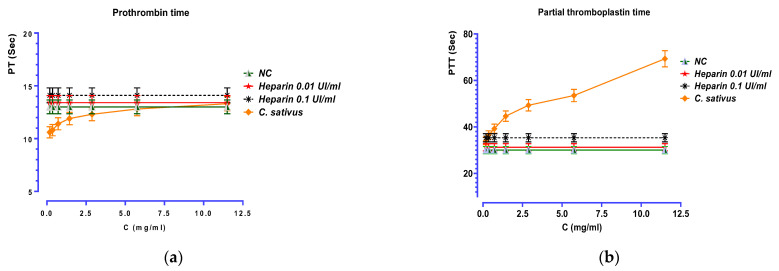
Effect of *C. sativus* decocted extract (E_0_), normal control (NC), and heparin on prothrombin time (**a**) and partial thromboplastin time (**b**).

**Figure 9 pharmaceuticals-16-00545-f009:**
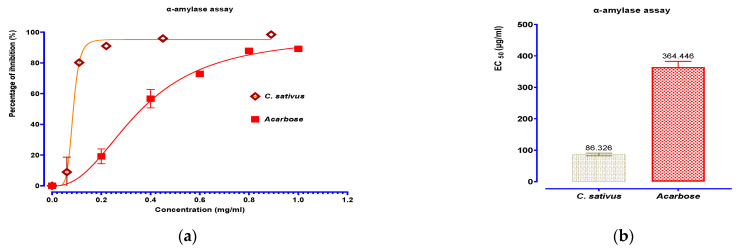
Percent inhibition and EC_50_ of inhibitory effects on α-amylase (**a**,**b**) and α-glucosidase (**c**,**d**) activities by *C. sativus* decocted extract and acarbose, in vitro. Values are means ± SEM (n = 3).

**Figure 10 pharmaceuticals-16-00545-f010:**
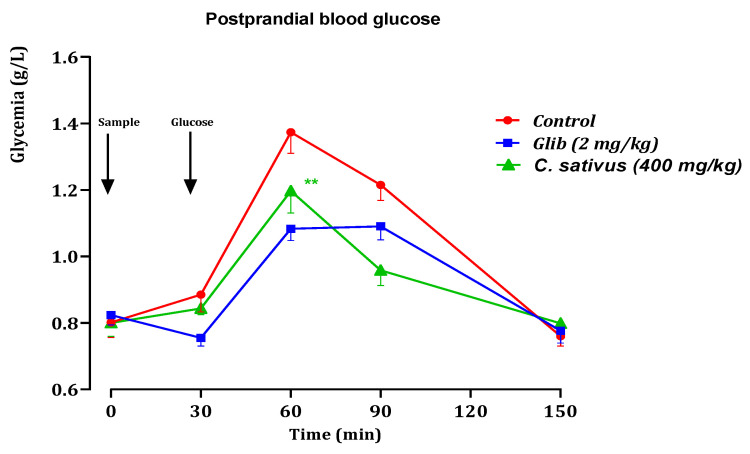
Variation in postprandial blood glucose in normal rats after administration of test products (decocted extract and glibenclamide). Values are means ± SEM. (n = 6). ** *p* < 0.01 in comparison with the control.

**Figure 11 pharmaceuticals-16-00545-f011:**
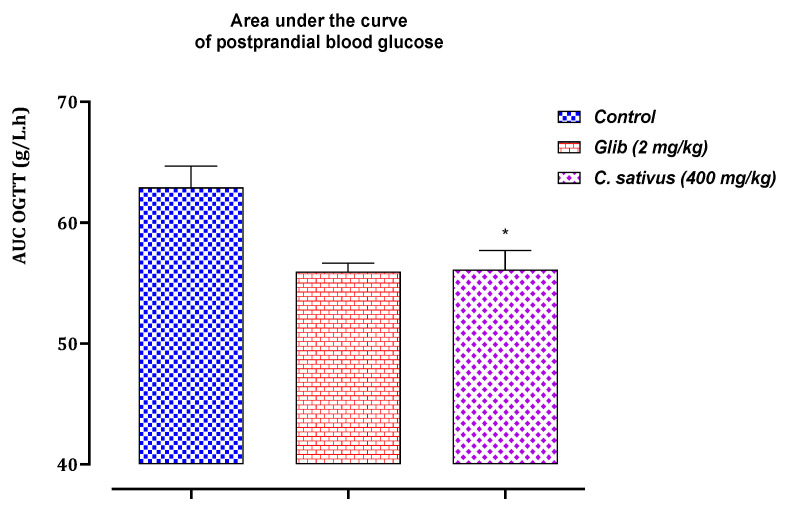
Variation in the area under the curve of postprandial blood glucose in normal rats after administration of tested products (decocted extract and glibenclamide). Values are means ± SEM. (n = 6). * *p* < 0.05 when compared with control.

**Table 1 pharmaceuticals-16-00545-t001:** The yield of the essential oil of *C. sativus*.

Species	Properties		
	Yield (%)	Color	Smell
*C. sativus*	0.25 ± 0.01	Deep yellow	Spicy and woody aromatics

**Table 2 pharmaceuticals-16-00545-t002:** Chemical profile (GC/MS) of the essential oil of *C. sativus*.

KI	Compounds	Area (%)
** 856 **	** (R)-(-)-2,2-dimethyl-1,3-dioxolane-4-methanol **	** 11.65 **
**912**	2-acetylfuran	1.42
**1017**	γ-terpinene	1.32
**1050**	2-acetylcyclopentanone	0.86
**1070**	*Trans*-arbusculone	0.91
**1121**	Isophorone	3.56
** 1095 **	** Phorone **	** 12.90 **
**1145**	**4-keto-isophorone**	**4.72**
** 1196 **	** Safranal **	** 6.39 **
**1230**	Duroquinone	1.10
**1252**	Thymoquinone	2.42
**1282**	2-ethyl menthone	0.93
**1289**	Cyclopent-2-en-1-one, 2-pentyl-	2.03
**1314**	2,3,4-trimethyl benzaldehyde	1.25
** 1436 **	** Dihydro-*β*-ionone **	** 8.62 **
**1455**	*β*-ionone epoxide	0.78
**1488**	***Trans*-*β*-ionone**	**4.81**
**1668**	(Z)-Coniferyl alcohol	3.62
**1800**	Octadecane	1.20
**1812**	Isopropyl myristate	2.38
**1900**	Nonadecane	1.91
**2000**	Eicosane	3.27
** 2010 **	** Isopropyl palmitate **	** 9.68 **
**2049**	4-tert-Octyl-*o*-cresol	1.29
**2060**	13-epi-manool	2.46
**2065**	**1-eicosanol**	**4.55**
**2100**	Heneicosane <n->	3.96
**Monoterpene hydrocarbons (%)**		**1.32**
**Oxygenated monoterpenes (%)**		**67.97**
**Hydrocarbon sesquiterpenes (%)**		**3.11**
**Oxygenated sesquiterpenes (%)**		**13.35**
**Hydrocarbon diterpenes (%)**		**7.23**
**Oxygenated diterpenes (%)**		**7.01**
**Total (%)**		**99.99**

**Table 3 pharmaceuticals-16-00545-t003:** Results of phytochemical tests.

Compounds/Species		*C. sativus*
Part used		*Stigma*
**Sterols and triterpenes**		+++
**Flavonoids**		++
**Anthocyanins**		-
**Tannins**	Catechic tannins	-
	Gallic tannins	+
**Anthracene derivatives**	Quinones	-
	O-Heterosides	-
	C-Heterosides	-
**Saponosides**		+
**Alkaloids**	Dragendorff	++
	Mayer	++

Category: Strong presence: +++; average presence: ++; low presence: + and absent: -.

**Table 4 pharmaceuticals-16-00545-t004:** List of compounds identified by mass spectrometry in the stigmas of *C. sativus*.

N° Pics	TR (min)	Molecules	Classes	MS [M-H]^−^ *m/z*	Exact Weights	Area (%)		
						E (0)	E (1)	E (2)
**1**	**4.04**	Safranal	Other	149	150	0.78	3.83	0.69
**2**	**4.28**	Caffeoyl coumaroyl methyl citric acid	Phenolic acid	551-387	552	0.76	2.08	0.59
**3**	**4.63**	Quinic malonyl glucoside acid	Other	439	440	0	0.17	0.07
**4**	**4.75**	Hydroxybenzoic acid hexoside	Polyphenol	299-239-179	300	0.01	0.29	0.02
**5**	**4.90**	Caffeic acid glycoside dimmer	Phenolic acid	683	684	0.02	0.34	0.03
**6**	**5.68**	Isorhamnetin 3-*O*-neohesperidoside	Flavonoide	623	624	0.01	0.15	0.02
**7**	**9.59**	Kaempferol 3-(6″-acetylglucoside)	Flavonoid	489	490	0.63	1.77	1.02
**8**	**14.16**	Syringetin hexoside	Flavonoid	345-507	508	5.86	3.09	2.8
**9**	**15.14**	Xanthoangelol	Flavonoid	391	392	1.57	2.77	1.3
**10**	**15.77**	Secoisolariciresinol	Polyphenol	361	362	2.52	0	1.53
**11**	**17.30**	Caffeic acid	Phenolic acid	179	180	0.01	0.13	0.01
**12**	**17.25**	Vanillic acid-dihexoside	Phenolic acid	537 [M + HCOOH-H]^−^	492	1.65	0	1.05
**13**	**17.66**	α-tocopherol	Other	429	430	0.66	0	0
**14**	**19.55**	Picrocrocin	Carotenoid	375 [M-H + HFA]^−^	330	18.78	30.23	0.5
**15**	**20.22**	Oleanolic acid	Fatty acid	409	456	4.26	16	6.85
**16**	**23.08**	Apigenin-*O*-rhamnoside	Flavonoid	415	416	1.06	0.52	0.64
**17**	**24.67**	Gallocatechin-pgd-3-*O*-glucoside	Flavonoid	737	738	1.63	1.1	3.68
**18**	**25.19**	*Trans*-crocin-4 (*trans*-crocetin di(b-*D*-gentiobiosyl) ester)	Carotenoid	1021 [M-H + HFA]^−^	976	10.52	1.32	17.99
**19**	**25.45**	*Trans*-crocin-3 (*trans*-crocetin (b-*D*-glucosyl)-(b-*D*-gentiobiosyl) ester)	Carotenoid	859 [M-H + HFA]^−^	814	25.65	10.1	3.58
**20**	**25.90**	Quercetin	Flavonoid	347 [M-H + HFA]^−^	301	11.1	2.18	4.12
**21**	**26.18**	*Trans*-crocin-2 (*trans*-crocetin (b-*D*-gentibiosyl) ester)	Carotenoid	651	652	7.8	11.25	24.58
**22**	**26.27**	*β*-Carotene	Carotenoid	581 [M-H + HFA]^−^	536	1.32	4.44	7
**23**	**26.69**	Coumaric acid-hexoside	Phenolic acid	325	326	0.8	0	0
**24**	**26.95**	*Cis*-crocin-4	Carotenoid	975	976	1.06	2.58	4.12
**25**	**27.15**	*Cis*-crocin-3	Carotenoid	813	814	0.32	1.64	1.48
**26**	**27.26**	*Trans*-crocin-1	Carotenoid	489	490	0.62	2.34	4.3
**27**	**27.57**	Crocetin	Carotenoid	327	328	0.57	0.16	2.47

**Table 5 pharmaceuticals-16-00545-t005:** Heavy metal concentration (mg/L) and maximum limits FAO/WHO (2009).

Species	Arsenic (As)	Cadmium (Cd)	Chromium (Cr)	Iron (Fe)	Lead (Pb)	Antimony (Sb)	Titanium (Ti)
*C. sativus*	0.013	≤0.001	0.005	0.513	0.305	0.001	0.017
*Maximum limits (FAO/WHO)*	1	0.3	2	20	3	1	-

**Table 6 pharmaceuticals-16-00545-t006:** The MIC, MBC, and MFC (µg/mL) of *C. sativus* decocted extract, and the MIC of antibiotics and antifungal agent.

*Microorganism*		*C. sativus*		*Antibiotics **					*Antifungals ^#^*
		*MIC*	*MBC or MFC*	*Gentamycin*	*Amoxicillin–Clavulanate*	*Vancomycin*	*Trimethoprim–Sulfamethoxazole*	*Penicillin G*	*Terbinafine*
** *GPC* **	** *S. epidermidis* **	*>5000*	*>5000*	2		>8	>4/76		
	** *S. aureus BLACT* **	*5000*	*5000*	<0.5		2	<10		
	** *S. aureus STAIML/MRS/mecA/HLMUP/BLACT* **	*>5000*	*>5000*	2		>8	>4/76		
	** *S. acidominimus* **	*5000*	*5000*	≤250		<0.5		0.03	
	** *S. group D* **	*>5000*	*>5000*	>1000		<0.5		0.13	
	** *S. agalactiae (B)* **	*5000*	*5000*	≤250		>4		0.06	
	** *S. porcinus* **	*>5000*	*>5000*	≤250		<0.5		0.06	
	** *E. faecalis* **	*2500*	*2500*	≤500		1	≤0.5/9.5		
	** *E. faecium* **	*>5000*	*>5000*	≤500		>4	>4/76		
** *GNB* **	** *A. baumannii* **	*600*	*1200*	≤1	≤2/2		≤1/19		
	** *E. coli* **	*>5000*	*>5000*	2	8/2		≤1/19		
	** *E. coli* ** ** *ESBL* **	*>5000*	*>5000*	2	>8/2		>4/76		
	** *E. aerogenes* **	*>5000*	*>5000*	≤1	8/2		≤1/19		
	** *E. cloacae* **	*>5000*	*>5000*	>4	>8/2		>4/76		
	** *C. koseri* **	*2500*	^2500^	<1	>8/2		<20		
	** *K. pneumoniae* **	*5000*	*5000*	≤1	≤2/2		≤1/19		
	** *P. mirabilis* **	*2500*	*2500*	2	≤2/2		>1/19		
	** *P. aeruginosa* **	*>5000*	*>5000*	2	>8/2		4/76		
	** *P. fluorescence* **	*2500*	*2500*	4	>8/2		4/76		
	** *P. putida* **	*>5000*	*>5000*	>4	>8/2		>4/76		
	** *S. marcescences* **	*5000*	*5000*	4	>8/2		>4/76		
	** *Sallemonella *sp.** **	*2500*	*2500*	>4	8/2		>4/76		
	** *Shigella *sp.** **	*600*	*1200*	>4	8/2		>4/76		
	** *Y. enterolitica* **	*2500*	*2500*	≤1	8/2		2/38		
** *Yeasts* **	** *C. albicans* **	>5000	>5000						12.500
	** *C. kefyr* **	2500	2500						25.000
	** *C. krusei* **	>5000	>5000						50.000
	** *C. parapsilosis* **	2500	2500						6.250
	** *C. tropicalis* **	>5000	>5000						12.500
	** *C. dubliniensis* **	>5000	>5000						3.125
	** *S. cerevisiae* **	>5000	>5000						3.125
** *Molds* **	** *A. niger* **	2500	2500						3.125

*: the MIC (µg/mL) of the antibiotics was determined by the BD Phoenix™ identification and antibiogram instrument; ^#^: the MIC (µg/mL) of terbinafine was determined on a microplate.

**Table 7 pharmaceuticals-16-00545-t007:** Origin, part used, location, and harvesting period of *Crocus sativus*.

Scientific Name	Part Collected	Type of Extract Used	Harvesting Area						
			Region	Province	Municipality	Latitude (x)	Longitude (y)	Altitude (m)	Collection Period
***Crocus sativus* L.**	**Stigma**	EO and extract	Fez-Meknes	Boulemane	Serghina	33°20′44″ N	4°24′11″ W	1496 m	September 2019

**Table 8 pharmaceuticals-16-00545-t008:** List of bacterial and fungal strains tested with their references.

Strains		Abbreviations	References
**Gram-positive cocci**	*Staphyloccocus epidermidis*	*S. epidermidis*	5994
	*Staphyloccocus aureus BLACT*	*S. aureus BLACT*	4IH2510
	*Staphyloccocus aureus STAIML/MRS/mecA/HLMUP/BLACT*	*S. aureus STAIML/MRS/mecA/HLMUP/BLACT*	2DT2220
	*Streptococcus acidominimus*	*S. acidominimus*	7DT2108
	*Streptococcus group D*	*S. group D*	3EU9286
	*Streptococcus agalactiae*	*S. agalactiae*	7DT1887
	*Streptococcus porcinus*	*S. porcinus*	2EU9285
	*Enterococcus faecalis*	*E. faecalis*	2CQ9355
	*Enterococcuss faecium*	*E. faecium*	13EU7181
**Gram-negative bacilli**	*Acinetobacter baumannii*	*A. baumannii*	7DT2404
	*Escherichia coli*	*E. coli*	3DT1938
	*Escherichia coli* *ESBL*	*E. coli ESBL*	2DT2057
	*Enterobacter aerogenes*	*E. aerogenes*	07CQ164
	*Enterobacter cloacae*	*E. cloacae*	02EV317
	*Citrobacter koseri*	*C. koseri*	3DT2151
	*Klebsiella pneumonie ssp. pneumonie*	*K. pneumonie*	3DT1823
	*Proteus mirabilis*	*P. mirabilis*	2DS5461
	*Pseudomonas aerogenosa*	*P. aerogenosa*	2DT2138
	*Pseudomonas fluorescence*	*P. fluorescence*	5442
	*Pseudomonas putida*	*P. putida*	2DT2140
	*Serratia marcescens*	*S. marcescens*	375BR6
	*Salmonella *sp.**	*Salmonella *sp.**	2CG5132
	*Shigella *sp.**	*Shigella *sp.**	7DS1513
	*Yersinia enterocolitica*	*Y. enterocolitica*	ATCC27729
**Yeasts**	*Candida albicans*	*C. albicans*	Ca
	*Candida kefyr*	*C. kefyr*	Cky
	*Candida krusei*	*C. krusei*	Ckr
	*Candida parapsilosis*	*C. parapsilosis*	Cpa
	*Candida tropicalis*	*C. tropicalis*	Ct
	*Candida dubliniensis*	*C. dubliniensis*	Cd
	*Saccharomyces cerevisiae*	*S. cerevisiae*	Sacc
**Fungi**	*Aspergillus niger*	*A. niger*	AspN

**Table 9 pharmaceuticals-16-00545-t009:** Extraction coding.

Extraction Methods	Solvents	Codification
**Soxhlet**	Ethanol/Water (70/30; *v*/*v*)	E (2)
	Water	E (1)
**Decoction**	Water	E (0)

## Data Availability

Not applicable.

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
