# Peer review of "Identification of Compounds of Crocus sativus by GC-MS and HPLC/UV-ESI-MS and Evaluation of Their Antioxidant, Antimicrobial, Anticoagulant, and Antidiabetic Properties"

_pharmaceuticals, 2023, doi:10.3390/ph16040545_

Round 1

Reviewer 1 Report

Decision:

Minor Revision

Comments

         The authors reported

This research article entitled Separation and identification of volatile and non-volatile compounds of Crocus sativus by GC-MS and HPLC/UV-ESI-MS3 and evaluation of their antioxidant, antimicrobial, anticoagulant and antidiabetic properties can be helpful in the scientific community after some revision. Overall, the work is good and well presented.  However, the authors should address the following points to improve scientific quality. After carefully addressing the suggested revisions, this work may be considered for publication.

1.                   First title is too lengthy short and make it more clear.

2.                   The abstract is too long. Highlight the importance of this work only and provide informative information only.  

3.                   In introduction section, cite more reference related to this work.

https://doi.org/10.3390/antiox11061205

https://www.mdpi.com/1420-3049/27/18/5935  

 4. Figure 5b is not clear. Provide high-quality images.

5. There are many grammatical errors, recheck the manuscript once again for all typo errors.

Reviewer 2 Report

The authors studied the volatile and non-volatile components of Crocus sativus and evaluated their biological activities. This manuscript is relatively systematic, but the writing of the article needs to be improved.

1. The Latin names of plants and bacteria appearing in abstract and text should be written in a standard way. When the names of plants of the same genus appear for the second time and later, the generic names need to be abbreviated. Please carefully check the Latin writing format of the full text.

2. In fig.5, 27 compounds were identified, while only 26 compounds were found in Fig. 5. 

3. It is better to give the chemical structures of the 27 identified compounds in a figure.

4. Fig.5, However, some peaks are too low, which looks like noise signals. I don't know the accuracy of the identification results.

5. Tab.4, the initial letters of the compound name need to be capitalized.

6. The format of references needs to be unified. For example, the first letter of each word in some quoted article titles is capitalized, which is wrong.

7. The writing of the MS should be carefully improved, since the full text appears too rough.

Reviewer 3 Report

pharmaceuticals-2205029 manuscript entitled "Liquid Chromatography in Analysis of Bioactive Compounds for Pharmaceuticals, Cosmetics, and Functional Food Interest" is not original as there are several works on this topic, and also on Moroccan saffron specifically.

In particular, I found a very similar one which I report below:

-Zaazaa L, Naceiri Mrabti H, Ed-Dra A, Bendahbia K, Hami H, Soulaymani A, Ibriz M. Determination of Mineral Composition and Phenolic Content and Investigation of Antioxidant, Antidiabetic, and Antibacterial Activities of Crocus sativus L. Aqueous Stigmas Extracts. Adv Pharmacol Pharm Sci. 2021 May 29;2021:7533938. doi: 10.1155/2021/7533938. PMID: 34195613; PMCID: PMC8181092

Furthermore, the manuscript suffers from very serious experimental and methodological shortcomings. The work is very confusing, it seems like a disjointed set of methods and activities without any logical sense.

The methods are not described adequately and in some cases they are not really explicit in the text. The materials and methods are written in bulk, first authors talk about the antibacterial activity, then about the EO and GC analysis, then about the phytochemical screening on the powder as it is.

In short, meaningless.

The authors should reorganize all the paper, to give the example of materials and methods, they should first illustrate well the characteristics, origin, identification and processing of the plant material, then talk about phytochemical screening since it concerns simply stigmas dried powder  by accurately describing the methods used, then move on to the description of the preparation of the essential oil and the extracts, to their phytochemical characterization and finally to their biological activity.

However, the methods as a whole, beyond the organization of the text, are written badly, in a deficient and inappropriate way.

I also have many doubts about the type of extraction used for polyphenols. Polyphenols are thermo and photosensitive, therefore an extraction in soxhelet is absolutely to be excluded. This could also explain the low biological activity found, because in my opinion most of them degrade during extraction!

Considering what mentioned above, I am forced to reject the manuscript.

Reviewer 4 Report

The article describes the investigation of C. sativus essential oils and organic extracts. The English is average and must be revised by a Native English speaker, additionally, we recommend the application of the following remarks to improve the quality of this paper:

- The title contain separation and identification, there is no separation or purification, and the word "separation" must be removed from the title. - Introduction is not well structured, it's better to focus on C. sativus, anticoagulant and antimicrobial activities. -  I suggest to segmented the paper  into two parts essential oils and organic extracts, since they are all extracts obtained from the same plant material. we advise you to merge the yields of all extracts. - The cultivar in this region has been introduced from the south of Morocco (TALIOUINE), so the chemical composition must be the same,  - Materials and methods must be cited before the results part - Add space between the values and the units. - There is no need to put the spectra in the article, it's better to move them to the annex section. - Figure 2 is inserted inside a table please rectify it - Table 2 has to be corrected - The equations should be inserted using the MS word automatic tool as requested by the MDPI guidelines - Correct the antidiabetic title in the material and methods section - References should be formatted to the MDPI format please check the MDPI guidelines  

Round 2

Reviewer 3 Report

I understand that authors need to publish their work, like all those who do research after all. However, this does not justify the publication of work that has been conducted inappropriately or that does not meet a minimum quality standard to be published in a particular journal, in this case Pharmaceuticals.

I am also willing to give the authors another opportunity to improve the manuscript, but all suggestions and comments must be resolved in order for me to reconsider it for publication, otherwise I am sorry but it cannot be accepted as it stands.

Given the current condition of the manuscript, I would have recommended a reject and encourage the resubmission, I don't have this opportunity so I choose major revision but I already point out that if the problems will be not resolved, my answer will be the same also at the next round of peer-review .

Best regards

Author Response

Answer

Thank you, dear professor, for your comments and for your interest in this work. We have tried to improve the manuscript according to your remarks and suggestions. We hope that this version will be perfect for publication.

Sincerely

Reviewer 4 Report

This paper is messy and devalorizes drastically the work it presents. The English is awful and must be revised by a Native English speaker, additionally, none of our previous corrections had been applied. This paper is very weak and needs major improvements. We recommend the application of the following remarks else we will strongly be against the publication of this paper:

1. the title mentions separation and identification, there is no separation or purification, and the word "separation" must be removed from the title. 2. the title should contain the biological activities that you tested ( not only the antidiabetic activity). 3. the abstract is messy, it must be structured (describe the methodology then the results and briefly interpret them) and check how the abstracts are written on similar papers.  4. introduction is not well structured, it's better to focus on C. sativus, anticoagulant and antimicrobial activities. 5. you need to add space between the values and the units. 6. you have segmented your article into two parts essential oils and organic extracts, since they are all extracts obtained from the same plant material we advise you to merge the yields of all extracts. 7. there is no need to put the spectra in the article, it's better to move them to the appendix section. 8. Figure 2 is inserted inside a table please rectify it 9. Table 2 has to be corrected 10. the phytochemical screening method is not clear, it should be revised 11. in line 832: the reference had not been inserted as recommended by MDPI guidelines "Chatsumpun and coworkers in 2017 is incorrect " 12. in lines 576, 730, 826, 843 : the equation has to be numbered and cited in the text. 13. the equations should be inserted using the MS word automatic tool as requested by the MDPI guidelines 14. correct the antidiabetic title in the material and methods section 15. references should be formatted to the MDPI format please check the MDPI guidelines

Author Response

Answer

Thank you, dear professor, for your comments and for your interest in this work. We have tried to improve the manuscript according to your remarks and suggestions. We hope that this version will be perfect for publication.

Cordially

Round 3

Reviewer 3 Report

The manuscript was improved according to comments and suggestions. However, I recommend before publication, an extensive language revision of the main text.

Best regards

Reviewer 4 Report

Most of our remarks and requests for modifications have not been taken into consideration.
